# The Synergistic Effect of MoS_2_ and NiS on the Electrical Properties of Iron Anodes for Ni-Fe Batteries

**DOI:** 10.3390/nano12193472

**Published:** 2022-10-04

**Authors:** Hongwei Tang, Mengyue Liu, Lingna Kong, Xiaoyan Wang, Yue Lei, Xige Li, Yan Hou, Kun Chang, Zhaorong Chang

**Affiliations:** 1Collaborative Innovation Center of Henan Province for Green Manufacturing of Fine Chemicals, Key Laboratory of Green Chemical Media and Reactions, Ministry of Education, School of Chemistry and Chemical Engineering, Henan Normal University, Xinxiang 453007, China; 2Henan Troily New Energy Technology Co., Ltd., Xinxiang 453000, China; 3Jiangsu Key Laboratory of Materials and Technology for Energy Conversion, College of Materials Science and Technology, Nanjing University of Aeronautics and Astronautics, Nanjing 210016, China

**Keywords:** nickel-iron battery, iron anode, Fe_3_O_4_, sulfide additives

## Abstract

In this paper, a series of Fe_3_O_4_/MoS_2_/NiS composite electrodes were synthesized by a simple coprecipitation method. The influence of different ratio additives (MoS_2_ and NiS) on the performance of iron anodes for Ni-Fe batteries was systematically investigated. In this paper, the mixed alkaline solution of 6 mol/L NaOH and 0.6 mol/L LiOH was used as electrolyte, and sintered Ni(OH)_2_ was used as counterelectrode. The experimental results show that the MoS_2_ and NiS additives can effectively eliminate the passivation phenomena in iron electrodes, reduce the electrode polarization, and increase the reversibility capacity. As a result, the Fe_3_O_4_/MoS_2_/NiS composite electrodes exhibit a high specific capacity, good rate performance, and long cycling stability. Especially, the Fe_3_O_4_/MoS_2_ (5%)/NiS (5%) electrode with a suitable ratio of additives can provide excellent electrochemical performance, with high discharge capacities of 657.9 mAh g^−1^, 639.8 mAh g^−1^, and 442.1 mAh g^−1^ at 600 mA g^−1^, 1200 mA g^−1^, and 2400 mA g^−1^, respectively. This electrode also exhibits good cycling stability.

## 1. Introduction

At present, battery systems powered by lithium-ion batteries and lead-acid batteries are widely used in various vehicles, communication base stations and energy storage fields [1,2,3]. Although lead-acid batteries exhibit stable performance and are low cost, their use has been gradually restricted by various countries because of lead pollution to the environment during the production process and use. Although lithium-ion batteries have a high working voltage and high energy density, the electrolyte contains ethylene carbonate-based organic solvents. When lithium-ion batteries are used as vehicle and large-scale energy storage power sources, their safety performance arouses widespread concern [4,5]. In comparison, alkaline batteries based on nickel-metal hydrides, nickel-iron and nickel-zinc are extremely safe due to the use of alkaline aqueous solutions as the electrolyte, are made from materials with abundant resources and less harmful effects on the environment. As power batteries, they are expected to be widely used in the fields of engineering vehicles, such as forklifts, ships, rail transportation, and large energy storage. In particular, nickel-iron batteries are fabricated from abundant and cheap iron resources, which are more competitive in the market [6,7,8,9,10]. In 1901, Edison and Junger invented the nickel-iron battery, which is a highly safe rechargeable battery that has a long service life and overcharge and discharge resistance and has been used in locomotive traction and compartment lighting [11,12,13]. However, the application of Fe-Ni batteries is greatly limited because iron electrodes are easily passivated, leading to a low discharge capacity and poor charge-discharge performance at low temperatures and large rates [14,15].

In view of the existing problems of iron electrodes, a large number of studies have been conducted on the active materials of iron electrodes [16,17,18,19], the electrode structure [14,20,21,22,23], additives [24,25,26], the surface coating of Fe_3_O_4_ [27,28], etc. In these studies, Bi_2_S_3_, PbS, FeS, NiS, Na_2_S, K_2_S, and other sulfide additives have significant effects, inhibiting the passivation of iron electrodes and increasing the utilization rate of the active materials in them. Posada et al. thoroughly studied the influence of Bi_2_S_3_, FeS, and K_2_S on nickel-iron batteries and proved that Bi_2_S_3_ and FeS additives are beneficial to the charging and discharging process of iron electrodes and that the synergistic effect of the elements Bi and S can inhibit the precipitation of hydrogen to a large extent, thereby improving the coulombic efficiency [29,30]. Wang et al. synthesized core-shell structured Fe_3_O_4_@NiS nanocomposites, and their experiments showed that the specific capacity of Fe_3_O_4_ nanocomposites coated with 5 wt.% NiS can reach 507 mAh g^−1^ at a low current [31]. Zhang et al. used nickel-pyridine xanthate to prepare a uniform NiS layer on the surfaces of Fe_3_O_4_ particles by the solution impregnation method, which greatly improved the activation rate and specific discharge capacity of the electrode [27]. It is obvious that metal sulfides can improve the electrochemical performance of iron electrodes. Zeng et al. prepared mesoscopic carbon/Fe/FeO/Fe_3_O_4_ hybrid materials by a solid-state reaction, in which the trivalent iron optimized system was favorable for the redox kinetics, while the carbon layer could effectively promote the charge transfer and inhibit the occurrence of self-discharge. Therefore, the composite anode exhibited a high specific capacity of 604 mAh g^−1^ at 1 A g^−1^ and high cycling stability [32]. Li et al. developed a NiCo//Fe battery by constructing a hierarchical core-shell structure with TiN@Fe_2_O_3_/CNTF as anode and NiCoP@NiCoP/CNTF as cathode, which effectively shorten the diffusion path and improved ion transport rate. The battery exhibited excellent electrochemical performance (with a capacity of 0.77 mAh cm^−2^ and an energy density of 265.2 mWh cm^−3^) [33]. Wu et al. prepared a core-shell structured C-Fe anode, which exhibited a high specific capacity of 208 mAh g^−1^ and a capacity retention of 93% after 2000 cycles at 4 A g^−1^) [34].

MoS_2_ has a layered structure similar to that of graphite; the physical, optical, electrical and catalytic properties of this nanostructure are similar to those of graphene; and it has a wide range of applications in the fields of mechanical lubrication, semiconductors and catalysis [35,36]. However, the effect of MoS_2_ as an additive in iron electrode materials on the performance of Fe-Ni batteries has not been reported. In this paper, we use MoS_2_ and NiS as additives in Fe_3_O_4_ anode materials to study their effects on the performance of iron electrodes. The mixed alkaline solution of 6 mol/L NaOH and 0.6 mol/L LiOH was used as the electrolyte, and sintered Ni(OH)_2_ was used as the counterelectrode. According to the experimental results, the addition of only MoS_2_ or NiS can improve the discharge capacity and cycling performance of iron electrodes to a certain extent, but the addition of both MoS_2_ and NiS in a given proportion can improve the electrochemical performance of iron electrodes to a greater extent, indicating that the two additives have a certain synergistic promotion effect. Compared with the pure Fe_3_O_4_, Fe_3_O_4_/MoS_2_, and Fe_3_O_4_/NiS electrodes, the Fe_3_O_4_/MoS_2_/NiS electrode exhibits an excellent specific discharge capacities of 657.9 mAh g^−1^, 639.8 mAh g^−1^, and 442.1 mAh g^−1^ at 600 mA g^−1^, 1200 mA g^−1^, and 2400 mA g^−1^, respectively. This electrode also exhibits good cycling stability.

## 2. Materials and Methods

### 2.1. Synthesis of Fe_3_O_4_ and NiS

The reagents used in the experiments were all analytical grade and were used without further purification. Fe_3_O_4_ was synthesized by chemical coprecipitation. The specific steps were as follows: 27.8 g of Fe_3_O_4_·6H_2_O (97%, Aladdin, Shanghai, China) and 7.5 g of NH_4_NO_3_ (98%, Macklin, Shanghai, China) were dissolved in 75 mL deionized water, and then the mixture was transferred to a three-necked flask. The reaction flask was placed in a water bath at 95 °C. Then, 10 mL of 2 mol/L NaOH (95%, Aladdin, Shanghai, China) solution was added dropwise until the pH reached about 13.5. This solution was continually stirred for 30 min. After the reaction was completed, the black precipitate was filtered and washed with deionized water several times at 70 °C until the liquor had no sulfate ions (no white precipitate was detected by barium chloride solution). After vacuum drying, the cleaned precipitated material was placed in a tubular furnace and calcined at 700 °C for 1 h under the protection of argon to obtain an Fe_3_O_4_ material with good crystallinity, which was ground through a 200-mesh sieve and used as a reserve.

The specific synthesis steps of NiS were as follows: 0.02 mol NiSO_4_·6H_2_O (AR, Aladdin, Shanghai, China) and 0.02 mol Na_2_S·9H_2_O (98%, Aladdin, Shanghai, China) are dissolved in 100 mL deionized water at the same time. The NiSO_4_ solution was added to a 500 mL reaction vessel, and the Na_2_S solution was slowly added under magnetic stirring at room temperature. After reaction for 1 h, the black precipitate was washed with deionized water several times, and finally, the target product was obtained in a vacuum drying oven at 60 °C.

The MoS_2_ sample provided by Henan Kelong Company (Xinxiang, China) was used as received without exfoliation.

### 2.2. Structural Characterization

The crystal structures of Fe_3_O_4_, NiS, MoS_2_, and their mixtures in certain proportions were characterized by polycrystalline X-ray diffraction (XRD, Bruker D8-A25 diffractometer using Cu Kα radiation (λ = 1.5406 Å Karlsruhe, Germany), and the micromorphologies of the samples were characterized by field emission scanning electron microscopy (FE-SEM, S-3400NII JEOL, Akishima, Japan).

### 2.3. Electrochemical Tests

Preparation of the iron electrode: The active materials Fe_3_O_4_, KS6 conductive graphite (KS6, the source of Taiyuan power, Taiyuan, China), polytetrafluoroethylene (D210C, the source of Taiyuan power, Taiyuan, China), and CMC (Battery grade, the source of Taiyuan power, Taiyuan, China) were mixed and ground in a mass ratio of 88:10:1:1. Then, an appropriate amount of highly pure water was added to form a uniformly mixed slurry. Finally, the slurry was evenly scraped onto nickel foam (1.5 cm × 1.5 cm), and dried at 80 °C for 6 h under vacuum conditions. The dried pole sheet was pressed into an approximately 0.5 mm electrode sheet under a pressure of 10 MPa, and the total load of each pole sheet was approximately 100 mg.

The Fe_3_O_4_/NiS/MoS_2_ electrode was prepared by adding NiS and MoS_2_ based on the percentage of the active substance. For example, if the addition amount of NiS and MoS_2_ was 5%, that is, Fe_3_O_4_:NiS:MoS_2_ was 90:5:5, then the electrode was labeled Fe_3_O_4_/NiS (5%)/MoS_2_ (5%), abbreviated Fe_3_O_4_/NiS/MoS_2_ (5%), and so on.

The analog battery was assembled with the structure of “one negative two positive”. A homemade iron electrode, commercially sintered Ni(OH)_2_ (Battery grade, Henan Kelong Company, Xinxiang, China), and a polypropylene film (Battery grade, Henan Kelong Company, Xinxiang, China) were used as the negative electrode, positive electrode and separator, respectively, both sides were clamped and fixed by an organic plastic plate; and the electrolyte was a mixed alkaline solution of 6 mol/L KOH (AR, Aladdin Shanghai, China) and 0.6 mol/L LiOH (98%, Aladdin, Shanghai, China).

A Wuhan Bluestone CT2001A (Wuhan, China) test equipment system was used to charge and discharge the batteries at a constant current and different current densities. Cyclic voltammetry (CV) and alternating current impedance (EIS) measurements were performed on a Shanghai Chenhua electrochemistry workstation (CHI660A, Shanghai, China). A three-electrode system was used in the test: the iron, nickel hydroxide and Hg/HgO electrodes were the working, counter, and reference electrodes, respectively, and the above-mentioned alkaline electrolyte was used. All electrochemical tests were conducted at 25 °C. The voltage window of the CV tests was −0.4–1.25 V, and the scanning rate was in the range of 1–8 mV/s. EIS was performed at frequencies between 1 mHz and 10 kHz.

## 3. Results and Discussion

### 3.1. Structural and Morphological Analyses of the Fe_3_O_4_, NiS, MoS_2_ and Fe_3_O_4_/MoS_2_/NiS (5%) Samples

Figure 1 shows the XRD patterns of the Fe_3_O_4_, NiS, MoS_2_, and Fe_3_O_4_/MoS_2_/NiS (5%) samples. As shown in Figure 1, all the diffraction peaks of Fe_3_O_4_ are consistent with the standard card PDF#75–0033, and the characteristic diffraction peaks are located at 30.12°, 35.47°, 43.11° and 62.59°, corresponding to the (220), (311), (400) and (440) crystal planes of Fe_3_O_4_, respectively. All the diffraction peaks of the MoS_2_ provided by the Henan Kelong Company (Xinxiang, China) correspond well to the standard card PDF#77–1716, with diffraction peaks at 14.40°, 32.69°, 39.56° and 49.81° corresponding to the (002), (100), (103) and (105) crystal planes of MoS_2_. The XRD pattern of the NiS products synthesized at room temperature has no obvious diffraction peaks, except three weak peaks at 2θ values of approximately 30.57°, 34.54° and 54.18°, indicating that amorphous NiS is generated. The XRD pattern of the Fe_3_O_4_/MoS_2_ (5%)/NiS (5%) mixed material clearly shows that the characteristic peaks of each material remain after mixing, indicating that Fe_3_O_4_ and the MoS_2_ and NiS additives are physically mixed, and no chemical reaction occurs between them.

Figure 2 shows field emission scanning electron microscopy images of the MoS_2_, NiS, Fe_3_O_4_ and Fe_3_O_4_/MoS_2_/NiS (5%) samples. The scanning electron microscopy images of MoS_2_ at different magnifications in Figure 2a,b indicate that MoS_2_ has a two-dimensional multilayer nanosheet structure. Figure 2c,d shows SEM images of NiS synthesized at room temperature at different magnifications. NiS has an irregular particle structure formed by the aggregation of many small crystal grains. As shown in Figure 2e, the Fe_3_O_4_ material is composed of spherical nanoparticles with a diameter of approximately 200 nm bonded together to form a multivoid particle structure. Figure 2f is the SEM image of the mixed sample of the three materials. The amorphous NiS particles and the layered MoS_2_ nanosheets are fused with the porous Fe_3_O_4_ through the physical grinding method, ensuring uniform mixing of the electrode materials.

Figure 3a shows the EDX images of Fe_3_O_4_/MoS_2_/NiS (5%) powder. Five elements, Fe, O, S, Ni and Mo, are present and evenly distributed. Figure 3b shows the EDX images of the Fe_3_O_4_/MoS_2_/NiS (5%) electrode sheet obtained by mixing with the PTFE binder. Figure 3b also shows the presence and even distribution of Fe, O, S, Ni, and Mo, indicating that all the elements are still evenly distributed during the preparation of the electrode sheet by slurry blending.

As shown in Appendix A, the MoS_2_ and NiS were successfully added in the Fe_3_O_4_ anode. The 2p_3/2_ peaks of Fe(III) and Fe(II) at 714.08 eV and 712.61 eV, and the corresponding 2p_1/2_ peaks at 727.56 eV and 725.72 eV are considered as the specific spectrum of Fe_3_O_4_ (Appendix A) [37]. The binding energy (BE) values for the most intense Mo 3d (Appendix A) and S 2p (Appendix A) doublets were found to be 230.80 eV (Mo 3d_5/2_) and 162.35 eV (S 2p_3/2_) [38]. These values correspond to MoS_2_ peaks reported in previous work. As shown in Appendix A, the binding energy at 857.1 and 874.65 eV represent the existence of Ni^2+^, which agrees well with the reported literature [39]. The XPS test results further show that MoS_2_ and NiS additives were successfully added to Fe_3_O_4_ anode.

### 3.2. Electrochemical Performance Analysis

Figure 4a–d show the charging-discharging curves of the Fe_3_O_4_, Fe_3_O_4_/MoS_2_, Fe_3_O_4_/NiS, and Fe_3_O_4_/MoS_2_/NiS electrodes (where MoS_2_:NiS = 1:1) with additive contents of 1.5%, 3.5%, 5%, and 10%, respectively, at a current density of 300 mA g^−1^. Figure 4e-h show the relationship between the discharge capacity of the iron electrode and the number of cycles at a current density of 300 mA g^−1^.

As shown in Figure 4a–d, the same trend is observed for each proportion of additives used, and two obvious discharge platforms corresponding to the oxidation process of the iron electrode (the electrode reaction processes of Fe/Fe^2+^ and Fe^2+^/Fe^3+^) appear. The discharge plateau voltage trend is Fe_3_O_4_ < Fe_3_O_4_/MoS_2_ < Fe_3_O_4_/NiS < Fe_3_O_4_/MoS_2_/NiS, and the discharge capacity trend is Fe_3_O_4_ < Fe_3_O_4_/MoS_2_ < Fe_3_O_4_/NiS < Fe_3_O_4_/MoS_2_/NiS. These results show that compared with the pure Fe_3_O_4_ electrode, the iron electrodes with MoS_2_ and NiS additives have higher discharge capacities, higher discharge plateau voltages and lower charging plateau voltages, which indicates that the internal resistance of the electrode reaction and the electrode polarization degree are reduced, and the reversibility of the electrode reaction is improved. For the Fe_3_O_4_/MoS_2_ electrode, this phenomenon may be related to the two-dimensional layer structure of MoS_2_. During the grinding process of the Fe_3_O_4_ active material and MoS_2_ additive, the peeling and lubricity of the MoS_2_ sheets enable the Fe_3_O_4_ particles to easily adhere to the lamella, which helps to increase the contact area with the active substance and is more conducive to charge transfer. Thus, the proton transfer rate and the discharging platform voltage increase, whereas the polarization degree of the electrode and the charging platform voltage decrease. Because NiS itself is an electrode material with good conductivity, adding a certain amount of NiS to the electrode material increases the conductivity of the electrode material, thereby reducing the internal resistance of the reaction and electrode polarization. When MoS_2_ and NiS additives are mixed with the active material at the same time, the synergistic promotion of the two additives further increases the conductivity of the electrode, reduces the ohmic polarization, and increases the reversibility of the electrode reaction, which is manifested in the charge and discharge platforms, i.e., the discharge platform is higher, whereas the charging platform is lower. A series of experimental comparisons reveal that when a certain amount of MoS_2_ or NiS is added to the electrode material, the discharge capacity and cycling stability of the electrode material are improved, and the electrode polarization is smaller than that of the Fe_3_O_4_ electrode. However, we find that Fe_3_O_4_ electrodes with both MoS_2_ and NiS as additives exhibit a higher discharge capacity and cycling stability and lower electrode polarization.

Figure 4e–h shows that whether only one additive (Fe_3_O_4_/MoS_2_ electrode or Fe_3_O_4_/NiS electrode) or two additives (Fe_3_O_4_/MoS_2_/NiS electrode, where MoS_2_:NiS = 1:1) are used, the discharge capacity is significantly improved relative to that of the pure Fe_3_O_4_ reference electrode; this phenomenon is especially obvious when a certain amount of only NiS is used or when two additives are used at the same time (MoS_2_:NiS = 1:1). An example is shown in Figure 4g where the amount of additive is 5%. At a current density of 300 mA g^−1^, the specific discharge capacity of the Fe_3_O_4_ electrode can reach 562.1 mAh g^−1^ in the first 40 cycles, and the capacity decreases in the subsequent cycles. After 100 cycles, the capacity retention rate is 75.7%. When the Fe_3_O_4_/MoS_2_ (5%) mixed material is used as the negative material, the specific discharge capacity can reach 609.9 mAh g^−1^ in 100 cycles, and the capacity retention rate can reach 88.1% after 100 cycles. The cause for the increased capacity may be related to the two-dimensional layered structure of MoS_2_; compared with the morphology of Fe_3_O_4_, which consists of spherical particles that aggregate to form secondary particles, the morphology of MoS_2_ is multilayer lamellar structure with a larger specific surface area, which can provide more chemical reaction sites and ion diffusion channels for the electrochemical reaction process and thus may explain the improvements in the specific discharge capacity and cycling stability. In addition, Figure 4g shows that the specific discharge capacity of the Fe_3_O_4_/NiS (5%) electrode can reach 631.4 mAh g^−1^ during cycling, and the capacity retention rate is 88.4% after 100 cycles. In this case, the increase in the specific discharge capacity of the iron electrode can be attributed to the combined effect of Ni^2+^ and S^2−^. In the process of charging and discharging, insoluble Fe(OH)_2_ is easily deposited on the surface of the iron electrode, which hinders the further reaction of internal Fe_3_O_4_ and reduces the utilization rate of the electrode material. However, the presence of S^2−^ can inhibit the formation of a passivation film on the surface of the iron electrode and the occurrence of the hydrogen precipitation reaction, and thus the utilization rate of the active materials and charging efficiency are improved to a certain extent. During the reaction process, Ni^2+^ may form Ni(OH)_2_, which has a similar crystal lattice to Fe(OH)_2_ and inhibits structural changes to a certain extent. The combined action of the two elements increases the specific discharge capacity and enhances the cycling performance of the electrode. Moreover, we find that when two additives are added at the same time in a ratio of 1:1 instead of alone as a single additive (Fe_3_O_4_/MoS_2_ electrode or Fe_3_O_4_/NiS electrode), the discharge capacity of the Fe_3_O_4_/MoS_2_/NiS electrode is higher during the charge and discharge process. For example, the discharge capacity of the Fe_3_O_4_/MoS_2_/NiS (5%) electrode can reach 638.7 mAh g^−1^, and after 100 cycles, it still retains a capacity of 584 mAh g^−1^. The electrode exhibits an excellent discharge capacity and cycling stability, which can be attributed to the synergistic effect of MoS_2_ and NiS.

The XRD and XPS results of Fe_3_O_4_/MoS_2_/NiS after cycling are provided in the Appendix A. Compared with the primary Fe_3_O_4_/MoS_2_/NiS (5%), the XRD peaks of the electrode after 100 cycles correspond to nickel foam and MoS_2_, respectively. No other new peaks are found, indicating that the crystal structure is stable after cycling. The full and high-resolution XPS spectrum also indicate that the composition of Fe_3_O_4_/MoS_2_/NiS (5%) electrode after cycling has not changed obviously. Therefore, both XRD and XPS results of the materials after cycling indicate that the sulfide additives can reduce the electrode polarization and improve the cycling stability. In conclusion, as directly shown in Figure 4, compared with the pure Fe_3_O_4_ reference electrode, the Fe_3_O_4_/MoS_2_ (5%), Fe_3_O_4_/NiS (5%) and Fe_3_O_4_/MoS_2_/NiS (5%) electrodes have significantly improved discharge capacities and cycling performances. Among them, the Fe_3_O_4_/MoS_2_/NiS (5%) electrode exhibits the highest specific discharge capacity and best cycling performance.

To study the effects of MoS_2_/NiS addition on the electrical properties of iron electrodes, Fe_3_O_4_, Fe_3_O_4_/MoS_2_/NiS (1.5%), Fe_3_O_4_/MoS_2_/NiS (3.5%), Fe_3_O_4_/MoS_2_/NiS (5%), and Fe_3_O_4_/MoS_2_(/NiS (10%) electrodes were prepared. Charge and discharge tests were conducted at current densities of 300 mA g^−1^, 600 mA g^−1^, 1200 mA g^−1^, and 2400 mA g^−1^. The experimental results are shown in Figure 5. As shown in Figure 5a–d, compared with the discharge capacity of the pure Fe_3_O_4_ electrode, the discharge capacity of the Fe_3_O_4_/MoS_2_/NiS electrode is significantly higher at both small and large current densities, and the discharge capacity gradually increases with increasing additive proportion.

When 5% of both MoS_2_ and NiS are added, the specific capacity is the highest; in particular, the discharge capacitance increases more obviously at a high current density. For example, at a current density of 600 mA g^−1^, the discharge capacities of the Fe_3_O_4_, Fe_3_O_4_/MoS_2_/NiS (1.5%), Fe_3_O_4_/MoS_2_/NiS (3.5%), Fe_3_O_4_/MoS_2_/NiS (5%), and Fe_3_O_4_/MoS_2_/NiS (10%) electrodes are 449.6 mAh g^−1^, 575.7 mAh g^−1^, 648.1 mAh g^−1^, 657.9 mAh g^−1^, and 621.2 mAh g^−1^, respectively. At a current density of 1200 mA g^−1^, the discharge capacities of the Fe_3_O_4_, Fe_3_O_4_/MoS_2_/NiS (1.5%), Fe_3_O_4_/MoS_2_/NiS (3.5%), Fe_3_O_4_/MoS_2_/NiS (5%), and Fe_3_O_4_/MoS_2_/NiS (10%) electrodes is 375.9 mAh g^−1^, 496.1 mAh g^−1^, 626.5 mAh g^−1^, 639.8 mAh g^−1^, and 624 mAh g^−1^, respectively. At a current density of 2400 mA g^−1^, the discharge capacity of the Fe_3_O_4_, Fe_3_O_4_/MoS_2_/NiS (1.5%), Fe_3_O_4_/MoS_2_/NiS (3.5%), Fe_3_O_4_/MoS_2_/NiS (5%), and Fe_3_O_4_/MoS_2_/NiS (10%) electrodes is 359.1 mAh g^−1^, 385.5 mAh g^−1^, 432.1 mAh g^−1^, 442.1 mAh g^−1^, 429.6 mAh g^−1^, respectively. Therefore, at different current densities, the trend in the specific discharge capacity of each electrode is Fe_3_O_4_/MoS_2_/NiS (5%) > Fe_3_O_4_/MoS_2_/NiS (3.5%) > Fe_3_O_4_/MoS_2_/NiS (10%) > Fe_3_O_4_/MoS_2_/NiS (1.5%) > Fe_3_O_4_. The discharge curves in Figure 5e–h show that the addition of sulfides in different proportions significantly increases the discharge platform voltage, reduces the charging platform voltage and the polarization degree of the electrode, and improves the reversibility of the reaction. In comparison, it is found that regardless of the current density, the Fe_3_O_4_/MoS_2_/NiS (5%) electrode has the highest discharge plateau, revealing excellent electrochemical performance. Therefore, according to the above experimental study, we find that when 5% MoS_2_ and NiS are added to the Fe_3_O_4_ electrode in a 1:1 ratio, the highest specific discharge capacity, smallest electrode polarization, and optimal electrode reaction reversibility are achieved.

To further study the influence of MoS_2_ and NiS additives on the electrical properties of iron electrodes, cyclic voltammetry measurements of the pure Fe_3_O_4_ electrode and Fe_3_O_4_/MoS_2_/NiS electrodes with different additive proportions were performed. Figure 6a–e show the CV curves of the Fe_3_O_4_, Fe_3_O_4_/MoS_2_/NiS (1.5%), Fe_3_O_4_/MoS_2_/NiS (3.5%), Fe_3_O_4_/MoS_2_/NiS (5%), and Fe_3_O_4_/MoS_2_/NiS (10%) electrodes at different scanning speeds. In Figure 6a–e, the CV curves of the five electrodes all show a pair of obvious redox peaks. With increasing scanning speed, the peak potential of the oxidation peak moves in a more positive direction and the peak potential of the reduction peak moves in a more negative direction. The potential difference increases gradually, which indicates that the polarization of each electrode increases with increasing current density. Through observation, we find that, at a given scanning speed, the degrees to which the redox peak current and the corresponding peak area increase are obviously different for the different sample electrodes. In comparison, the addition of two kinds of sulfide significantly affects the redox behavior of Fe_3_O_4_ electrodes. The peak currents of the Fe_3_O_4_/MoS_2_/NiS electrodes are always higher than that of the pure Fe_3_O_4_ electrode, indicating that the addition of a sulfide additive helps to reduce the charge and ion transfer resistances and increases the conductivity of the electrode, thereby improving the high-current discharge performance of the electrode.

Figure 6f shows the relationship between the anodic peak current and the square root of the scanning rate. As shown in Figure 6f, at a given scanning speed, the five sample electrodes have different peak currents. Among them, the Fe_3_O_4_/MoS_2_/NiS (5%) electrode has the largest peak current, indicating that it has a faster electrode reaction rate. According to Figure 6f, the fitted slopes of the five sample electrodes are 0.00528, 0.01124, 0.01187, 0.01625, and 0.0117. The magnitude of the slope can be used to indicate the magnitude of the diffusion coefficient. Based on the fitted data, the slope of the Fe_3_O_4_/MoS_2_/NiS (5%) electrode is clearly the largest, and the corresponding diffusion coefficient calculated according to the Randles–Sevcik equation is the largest, indicating that the increase in the transfer speed of OH^−^ helps to increase the reaction speed of the electrode. These results are consistent with the aforementioned charging-discharging trend.

Table 1 lists the cathodic and anodic peak potentials of the five electrodes at a scanning speed of 2 mV/s, where E_O_ and E_R_ represent the anode and cathode peak potentials, respectively, and △(E_O-R_) represents the difference between the anode and cathode peak potentials, which reflects the reversibility of the reaction. The smaller the value of △(E_O-R_) is, the better the reversibility of the reaction is. According to the table, the △(E_O-R_) value of the electrode decreases with increasing addition amount of MoS_2_ and NiS, and the △(E_O-R_) value of the Fe_3_O_4_/MoS_2_/NiS (5%) electrode is the smallest, indicating that the reaction reversibility of the Fe_3_O_4_/MoS_2_/NiS electrode is the best at this ratio. For the electrode with 10% addition, the increase in the △(E_O-R_) value may be related to the decrease in the Fe_3_O_4_ active substance.

Figure 7 shows the EIS profiles of the Fe_3_O_4_ electrode and Fe_3_O_4_/MoS_2_/NiS electrodes with different proportions of additives at open-circuit voltage. EIS was performed at frequencies between 1 mHz and 10 kHz. Here, Rs represents the total resistance of the electrolyte; Rc1 and Rc2 are the charge transfer resistances of the iron electrode and Ni(OH)_2_ counter electrode, respectively; CPE represents the electric double layer capacitance and W represents the Warburg impedance. The charge transfer resistance of the electrode is an important parameter for measuring the speed of the electrode reaction. A large charge transfer resistance indicates that the electrode reaction resistance is large, which is not conducive to the electrode reaction. In contrast, the smaller the charge transfer resistance is, the stronger the depolarization ability of the electrode reaction is. According to the curve fitted by the equivalent circuit, the trend in the charge transfer resistances of the five sample electrodes is: Fe_3_O_4_ (1.064 Ω cm^2^) > Fe_3_O_4_/MoS_2_/NiS (1.5%) (0.5454 Ω cm^2^) > Fe_3_O_4_/MoS_2_/NiS (10%) (0.5397 Ω cm^2^) > Fe_3_O_4_/MoS_2_/NiS (3.5%) (0.4638 Ω cm^2^) > Fe_3_O_4_/MoS_2_/NiS (5%) (0.4328 Ω cm^2^). It can be seen that adding a certain amount of MoS_2_ and NiS can effectively reduce the charge transfer resistance, which is beneficial to the electrode reaction. The decrease in the charge transfer resistance may be attributed to the fact that S^2−^ in MoS_2_ and NiS can adsorb on the surface of the iron electrode, inhibiting the formation of a passivation film on it. The increase in the contact surface between MoS_2_ and Fe_3_O_4_ is beneficial for decreasing the polarization of the electrode, and NiS with its good conductivity is beneficial for increasing the conductivity of the iron electrode. In addition, the synergistic effect of the two sulfides on the iron electrode results in the best electrochemical performance.

In the above study, the ratio of MoS_2_:NiS was fixed at 1:1, and the effects of different addition amounts on the performance of iron electrodes were investigated. To determine the optimal addition ratio of MoS_2_ and NiS, we used Fe_3_O_4_/MoS_2_/NiS (5%) as the comparison standard and changed the addition ratio of MoS_2_ and NiS to prepare Fe_3_O_4_/MoS_2_ (10%)/NiS (5%) and Fe_3_O_4_/MoS_2_ (5%)/NiS (10%) electrodes. Figure 8 shows the cycle capacity graphs and charging-discharging curves of the Fe_3_O_4_/MoS_2_/NiS (5%), Fe_3_O_4_/MoS_2_ (10%)/NiS (5%), and Fe_3_O_4_/MoS_2_ (5%)/NiS (10%) electrodes at current densities of 600 mA g^−1^, 1200 mA g^−1^, and 2400 mA g^−1^. As shown in Figure 8a,c,e, the Fe_3_O_4_/MoS_2_/NiS (5%) electrode has the highest discharge capacity and the best cycling stability at different current densities and exhibits a relatively excellent electrochemical performance at high current densities in particular. Similarly, Figure 8b,d,f shows that, for the same number of cycles, the Fe_3_O_4_/MoS_2_/NiS (5%) electrode has the largest discharge capacity, a higher discharge voltage plateau and a lower charging platform, that is, it exhibits the smallest electrode polarization, indicating that the reaction has the best reversibility. In summary, after a series of experiments, we conclude that when the additive ratio of MoS_2_:NiS is 1:1 and the addition amount is 5%, the discharge capacity of nickel-iron batteries is the highest, the electrode polarization is the smallest, and the battery reaction reversibility is the best.

Under the same conditions, cyclic voltammetry tests were performed for the Fe_3_O_4_/MoS_2_ (5%)/NiS (5%), Fe_3_O_4_/MoS_2_ (10%)/NiS (5%) and Fe_3_O_4_/MoS_2_ (5%)/NiS (10%) electrodes, and the test results are shown in Appendix A. The CV curves of the three electrodes have similar shapes with a pair of obvious redox peaks. With increasing scanning speed, the oxidation peak shifts to the right and the reduction peak shifts to the left. Appendix A shows the peak potentials and potential differences between the anode and cathode of the three sample electrodes at a scanning speed of 1 mV/s (see Appendix A). △(E_O-R_) represents the difference between the anode peak potential (E_O_) and cathode peak potential (E_R_). The smaller △(E_O-R_) is, the better the reversibility of the electrode reaction is. Appendix A shows that the reversibility of the electrode reaction is as follows: Fe_3_O_4_/MoS_2_ (5%)/NiS (5%) > Fe_3_O_4_/MoS_2_ (5%)/NiS (10%) > Fe_3_O_4_/MoS_2_ (10%)/NiS (5%), that is, the Fe_3_O_4_/MoS_2_ (5%)/NiS (5%) electrode has the best reversibility in the process of the charging-discharging reaction, which is consistent with the results of the aforementioned charging-discharging curves. Appendix A shows the relationship between the anodic peak current and the square root of the scanning rate. As shown in Appendix A, at different scanning speeds, each sample electrode has a different peak current, and at a given scanning speed, the Fe_3_O_4_/MoS_2_ (5%)/NiS (5%) electrode has the highest peak current, revealing its better conductivity and faster electrode reaction rate. According to Appendix A, the fitted slopes of the three sample electrodes are 0.01784, 0.01755, and 0.01607. It can be inferred from these fitting results that the Fe_3_O_4_/MoS_2_ (5%)/NiS (5%) electrode has the largest diffusion coefficient, which is more conductive to improving the reaction speed.

For the above three sample electrodes, we also conducted AC impedance tests, and the test results are shown Appendix A. According to the curve fitted by the equivalent circuit, the relationship between the charge transfer resistances of the three electrodes can be clearly observed, i.e., Fe_3_O_4_/MoS_2_ (5%)/NiS (5%) (0.4328 Ω cm^2^) < Fe_3_O_4_/MoS_2_ (5%)/NiS (10%) (0.4799 Ω cm^2^) < Fe_3_O_4_/MoS_2_ (10%)/NiS (5%) (0.5514 Ω cm^2^).

In summary, according to the above electrochemical analysis, it can be seen that the optimal ratio of NiS and MoS_2_ added to the Fe_3_O_4_ electrode is 5% of the mass of the active substance. At this ratio, the electrode has a small electrode transfer charge, better electrode reaction reversibility, and a larger diffusion coefficient, resulting in excellent electrochemical performance.

## 4. Conclusions

Sulfide additives play a very important role in improving the electrical properties of iron electrodes by effectively inhibiting the passivation of the iron negative electrodes and increasing their discharge capacity. In this paper, a simple coprecipitation method was used to prepare the Fe_3_O_4_ active material, and Fe_3_O_4_, MoS_2_, and NiS were mixed in different proportions by physical mixing and grinding methods to prepare iron anodes. The XRD and SEM results show that Fe_3_O_4_ is evenly mixed with NiS and MoS_2_, and the elements are evenly distributed. The results of constant current charge-discharge measurements show that the iron electrode with NiS and MoS_2_, which play a positive role in preventing the passivation of the iron electrode and in reducing its polarization degree, exhibits excellent electrochemical performance. In particular, when both MoS_2_ and NiS are mixed with the iron electrode as additives, their synergistic effect results in an iron electrode with better electrical properties. Among these electrodes, the Fe_3_O_4_/MoS_2_ (5%)/NiS (5%) mixed material electrode has the best electrical properties. At current densities of 600 mA g^−1^, 1200 mA g^−1^, and 2400 mA g^−1^, the specific discharge capacity can reach 657.9 mAh g^−1^, 639.8 mAh g^−1^, and 442.1 mAh g^−1^, and after 100 cycles of charging and discharging, the capacity retention rate can reach 88.5%, 84.9%, and 78.6%, respectively, showing excellent rate performance. Cyclic voltammetry and AC impedance analysis results shows that doping with NiS and MoS_2_ significantly improves the reversibility of the electrode reaction, increases the conductivity of the electrode and the ion diffusion coefficient, and reduces the charge diffusion resistance.

## Figures and Tables

**Figure 1 nanomaterials-12-03472-f001:**
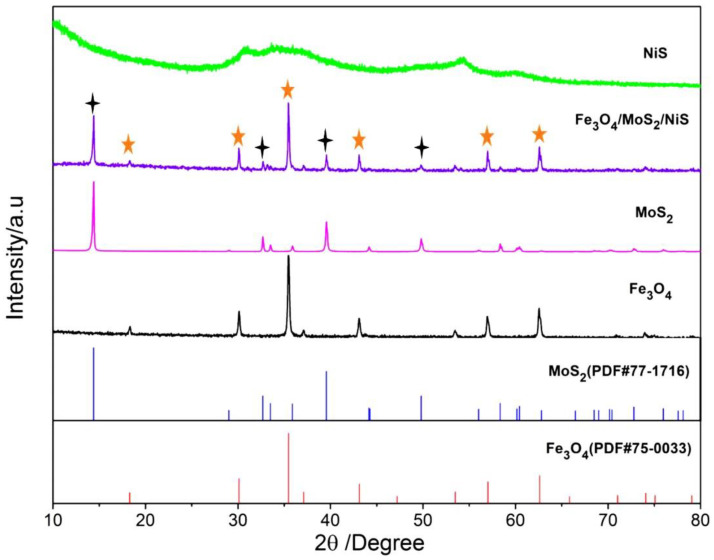
XRD patterns of the Fe_3_O_4_, NiS, MoS_2_ and Fe_3_O_4_/MoS_2_/NiS (5%) samples. (The black asterisk in the figure represents MoS_2_, and the orange asterisk represents Fe_3_O_4_).

**Figure 2 nanomaterials-12-03472-f002:**
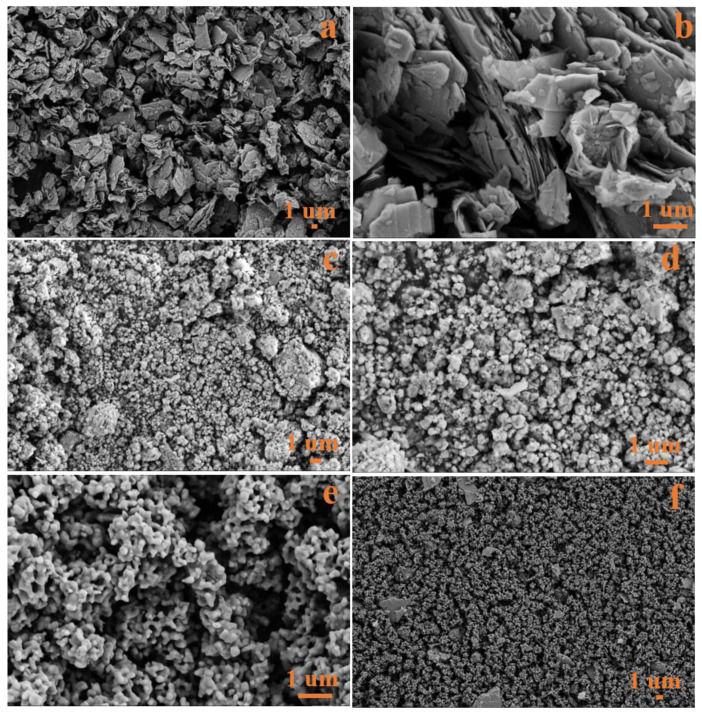
SEM images of the MoS_2_ (**a**,**b**), NiS (**c**,**d**), Fe_3_O_4_ (**e**) and Fe_3_O_4_/MoS_2_/NiS (5%) (**f**) samples.

**Figure 3 nanomaterials-12-03472-f003:**
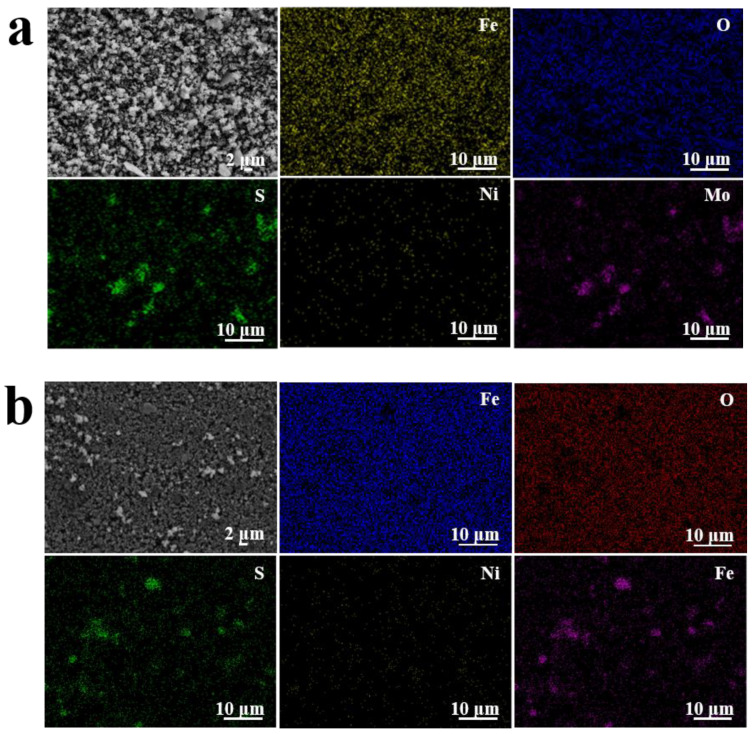
EDX images of the Fe_3_O_4_/MoS_2_/NiS (5%) powder (**a**) and the electrode sheet (**b**).

**Figure 4 nanomaterials-12-03472-f004:**
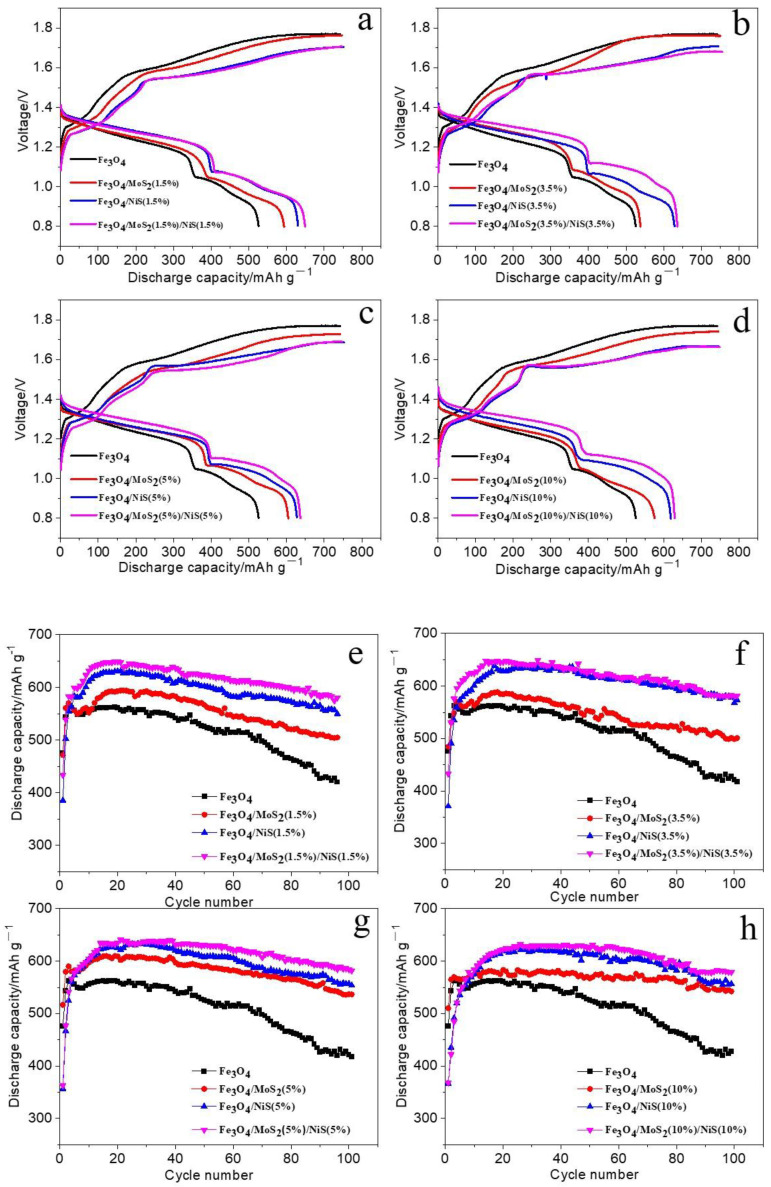
Charge-discharge curves and cycle capacity plots of, Fe_3_O_4_, Fe_3_O_4_/MoS_2_, Fe_3_O_4_/NiS and Fe_3_O_4_/MoS_2_/NiS electrodes at the current density of 300mA/g, when the additive content is 1.5% (**a,e**), 3.5% (**b,f**), 5% (**c,g**) and 10% (**d,h**) respectively.

**Figure 5 nanomaterials-12-03472-f005:**
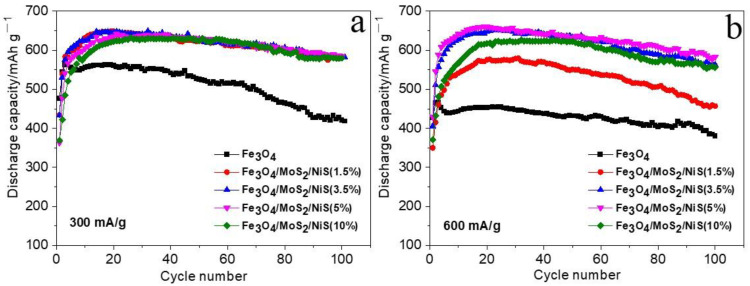
Cycle capacity plots and charging-discharging curves of Fe_3_O_4_, Fe_3_O_4_/MoS_2_/NiS (1.5%), Fe_3_O_4_/MoS_2_/NiS (3.5%), Fe_3_O_4_/MoS_2_/NiS (5%) and Fe_3_O_4_/MoS_2_/NiS (10%) electrodes at current density of 300 mA/g (**a,e**), 600 mA/g (**b,f**), 1200 mA/g (**c**) and 2400 mA/g (**d**) respectively.

**Figure 6 nanomaterials-12-03472-f006:**
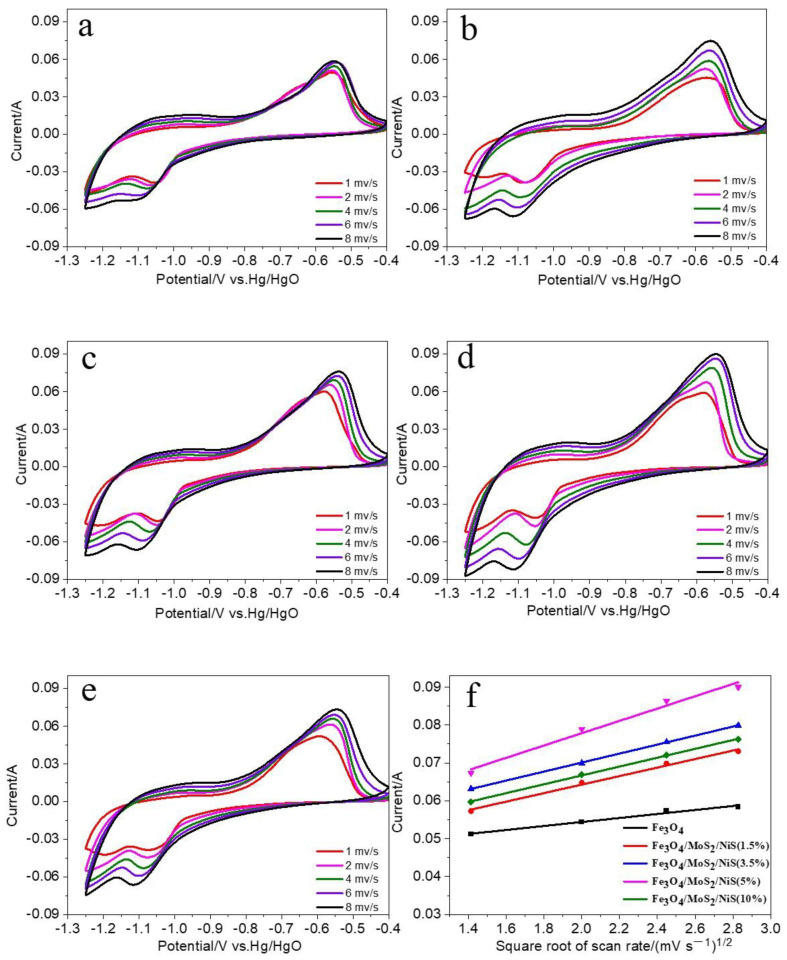
(**a**–**e**) Cyclic voltammetry curves of the Fe_3_O_4_, Fe_3_O_4_/MoS_2_/NiS (1.5%), Fe_3_O_4_/MoS_2_/NiS (3.5%), Fe_3_O_4_/MoS_2_/NiS (5%), and Fe_3_O_4_/MoS_2_/NiS (10%) electrodes; (**f**) Plots of the relationships between the maximum oxidation peak currents and the square root of the scanning speeds.

**Figure 7 nanomaterials-12-03472-f007:**
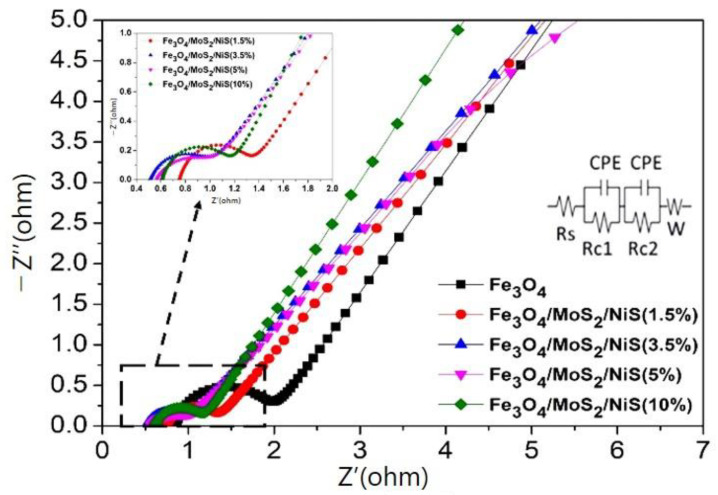
AC impedance spectra of the Fe_3_O_4_ electrode and Fe_3_O_4_/MoS_2_/NiS electrodes with different proportions of sulfides.

**Figure 8 nanomaterials-12-03472-f008:**
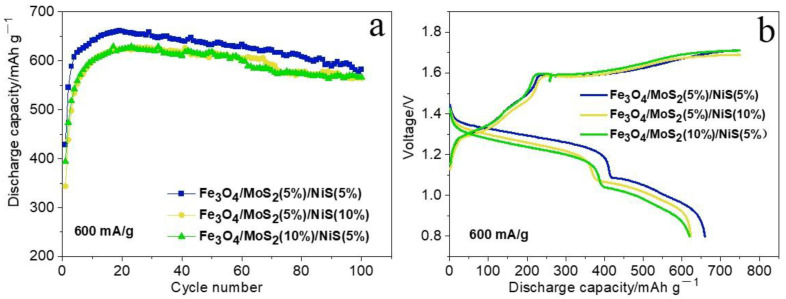
Cycling performance and charging-discharging curves of three electrodes at current densities of 600 mA g^−1^(**a**,**b**), 1200 mA g^−1^(**c**,**d**), and 2400 mA g^−1^(**e**,**f**).

**Table 1 nanomaterials-12-03472-t001:** Anode and cathode potentials and potential differences of five sample electrodes.

Electrode	E_O_ (mV)	E_R_ (mV)	△E_O-R_ (mV)
Fe_3_O_4_	554	1067	513
Fe_3_O_4_/MoS_2_/NiS (1.5%)	571	1079	508
Fe_3_O_4_/MoS_2_/NiS (3.5%)	562	1049	487
Fe_3_O_4_/MoS_2_/NiS (5%)	573	1053	480
Fe_3_O_4_/MoS_2_/NiS (10%)	565	1074	500

## Data Availability

The data presented in this study are available on request from the corresponding author.

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
