# Peer review of "The Synergistic Effect of MoS2 and NiS on the Electrical Properties of Iron Anodes for Ni-Fe Batteries"

_nanomaterials, 2022, doi:10.3390/nano12193472_

Round 1

Reviewer 1 Report

I recommend the publication of the manuscript "The influence of the synergistic effect of MoS2 and NiS on the electrocal properties of iron anodes" in Nanomaterials, after some minor revisions.

1) The type of battery used should be specified both in the TITLE and in the ABSTRACT.

2) At the end of the abstract the main results are summarized. The electrolyte and the counterelectrode should be mentioned as well.

3) INTRODUCTION, page 1, line 36-37: the sentence "do not pollute the environment" is misleading, since there isn't any technology with zero impact on the environment. Please, rectify it.

4) In the last of the INTRODUCTION, the electrolyte, the counterelectrode and the electrochemical performance (with number) should be briefly mentioned.

5) In MATERIALS AND METHODS specify the provider of Fe3O4, NH4NO3, NaOH, Na2S, NiSO4, graphite, PTFE, Ni(OH)2, CMC and polypropylene films, since only the provider of MoS2 is mentioned.

6) In MATERIALS AND METHODS specify the amount of NH4NO3 added in distilled water. "Certain amount" is too generic.

7) in MATERIALS AND METHODS specify the amount of NaOH 2M added to the solution and/or the pH reached. 

8) The synthesis of NiS involved the preparation of two solution (NiSO4 and Na2S). Please,specify whether 100 ml is the volume of each solution, since it is not clear. It seem that there is just one solution (100 ml) with both salts.

9) In STRUCTURAL CHARACTERIZATION specify the kind of diffractometer used.

10) in ELECTROCHEMICAL TESTS specify the amplitude of EIS. 

11) RESULTS AND DISCUSSION, page 8, line 259: there is a typo. Fe3O4/MoS2(/NiS (10%): please, remove the bracket. 

Author Response

I recommend the publication of the manuscript "The influence of the synergistic effect of MoS2 and NiS on the electrocal properties of iron anodes" in Nanomaterials, after some minor revisions.

Response: Thank you very much for your time and efforts on reviewing our paper. Your constructive comments are very important and helpful for further improving our manuscript.

  1. The type of battery used should be specified both in the TITLE and in the ABSTRACT.

Response: Thanks very much for your good suggestion. In the manuscript, we updated the title as "The synergistic effect of MoS2 and NiS on the electrical properties of iron anodes for Ni-Fe batteries". And the type of battery was also added in the introduction (1 page, 17 lines).

  1. At the end of the abstract the main results are summarized. The electrolyte and the counterelectrode should be mentioned as well.

Response: Thanks for the referee’s comments. In this paper, the mixed alkaline solution of 6 mol/L NaOH and 0.6 mol/L LiOH was used as the electrolyte, and sintered Ni(OH)2 was used as the counterelectrode. It has been complemented in the abstract.

  1. INTRODUCTION, page 1, line 36-37: the sentence "do not pollute the environment" is misleading, since there isn't any technology with zero impact on the environment. Please, rectify it.lessharmful effects on the environment

Response: According to your kind suggestion, the sentence was revised as “In comparison, alkaline batteries based on nickel-metal hydrides, nickel-iron and nickel-zinc are extremely safe due to the use of alkaline aqueous solutions as the electrolyte, are made from materials with abundant resources and less harmful effects on the environment.” It has been revised in the manuscript.

  1. In the last of the INTRODUCTION, the electrolyte, the counterelectrode and the electrochemical performance (with number) should be briefly mentioned.

Response: We are very grateful for your kind suggestion. Here, the mixed alkaline solution of 6 mol/L NaOH and 0.6 mol/L LiOH was used as the electrolyte, and sintered Ni(OH)2 was used as the counterelectrode. Compared with the pure Fe3O4, Fe3O4/MoS2 and Fe3O4/NiS electrodes, the Fe3O4/MoS2/NiS electrode exhibits an excellent specific discharge capacities of 657.9 mAh g-1, 639.8 mAh g-1 and 442.1 mAh g-1 at 600 mA g-1, 1200 mA g-1 and 2400 mA g-1, respectively. This electrode also exhibits good cycling stability. It has been complemented in the INTRODUCTION.

  1. In MATERIALS AND METHODS specify the provider of Fe3O4, NH4NO3, NaOH, Na2S, NiSO4, graphite, PTFE, Ni(OH)2, CMC and polypropylene films, since only the provider of MoS2is mentioned.

Response: We thank the referee’s suggestion. Fe3O4 was synthesized by chemical coprecipitation. And the providers of NH4NO3, NaOH, Na2S, NiSO4, graphite, PTFE, Ni(OH)2, CMC and polypropylene films have been complemented in MATERIALS AND METHODS.

  1. In MATERIALS AND METHODS specify the amount of NH4NO3added in distilled water. "Certain amount" is too generic.

Response: We are very grateful for your kind suggestion. Here, 27.8 g Fe3O4·6H2O (97%, Aladdin) and 7.5g NH4NO3 (98%, Macklin) were dissolved in 75 mL deionized water. It has been complemented in MATERIALS AND METHODS.

  1. In MATERIALS AND METHODS specify the amount of NaOH 2M added to the solution and/or the pH reached.

Response: We are very grateful for your kind suggestion. The reaction flask was placed in a water bath at 95 oC, then 10 ml of 2 mol/L NaOH (95%, Aladdin) solution was added dropwise until the pH reached about 13.5. It has been complemented in MATERIALS AND METHODS.

  1. The synthesis of NiS involved the preparation of two solution (NiSO4and Na2S). Please,specify whether 100 ml is the volume of each solution, since it is not clear. It seem that there is just one solution (100 ml) with both salts.

Response: We are very grateful for your kind suggestion. Both of the two salts are dissolved in 100 ml deionized water at the same time. It has been revised in the manuscript.

  1. In STRUCTURAL CHARACTERIZATION specify the kind of diffractometer used.

Response: We gratefully appreciate the very careful reviews.The crystal structures of the compouds were characterized by polycrystalline X-ray diffraction (XRD, Bruker D8-A25 diffractometer using Cu Kα radiation (λ=1.5406 Å). It has been complemented in STRUCTURAL CHARACTERIZATION.

  1. In ELECTROCHEMICAL TESTS specify the amplitude of EIS.

Response: Thanks very much for your good suggestion. The amplitude of EIS is 1 mHz and 10 kHz. It has been complemented in the manuscript.

  1. RESULTS AND DISCUSSION, page 8, line 259: there is a typo. Fe3O4/MoS2(/NiS (10%): please, remove the bracket.

Response: We are sorry for our inattention. We have corrected the typos throughout the whole manuscript as far as we can.

Reviewer 2 Report

Authors have discussed the synergistic effect of MoS2 and NiS with iron electrodes. Authors have done the structural and electrochemical analysis of prepared pure and doped iron oxides. However, the following revision work is needed for the publication.

1.     Authors should be elaborated the recent outcomes of Ni-Fe batteries in the introduction.

2.     CVs are showing the similar trend of Fe3O4 in the Fig.6, why? What is the role of MoS2 and NiS?

3.     Figure 8 caption is not clear.

4.     Authors should be provided the EDX and XPS results for the compositional confirmation.

5.     How authors confirmed the hindrance of Fe3O4 passivation after the inclusion of MoS2 and NiS?

6.     Authors are suggested to provide the structural and/or surface analysis after the stability test.

7.     Title of the manuscript is not clear. Should be updated the title appropriately.

Author Response

  1. Authors should be elaborated the recent outcomes of Ni-Fe batteries in the introduction. 

Response: Thank you so much for your suggestion. In the revised manuscript, we have complemented the recent literatures on Ni-Fe batteries in the introduction. Zeng et al. prepared mesoscopic carbon/Fe/FeO/Fe3O4 hybrid materials by solid-state reaction, in which the trivalent iron optimized system was favorable for the redox kinetics, while the carbon layer could effectively promote the charge transfer and inhibit the occurrence of self-discharge. Therefore, the composite anode exhibited a high specific capacity of 604 mAh g-1 at 1 A g-1 and high cycling stability [32]. Li et al. developed a NiCo//Fe battery by constructing a hierarchical core-shell structure with TiN@Fe2O3/CNTF as anode and NiCoP@NiCoP/CNTF as cathode, which effectively shorten the diffusion path and improved ion transport rate. The battery exhibited excellent electrochemical performance (with a capacity of 0.77 mAh cm-2 and an energy density of 265.2 mWh cm-3) [33]. Wu et al. prepared a core-shell structured C-Fe anode, which exhibited a high specific capacity of 208 mAh g-1 and a capacity retention of 93% after 2000 cycles at 4 A g-1) [34]. It has been complemented in the manuscript.

  1. CVs are showing the similar trend of Fe3O4in the Figure6, why? What is the role of MoS2 and NiS?

Response: We thank the referee’s comments. The CV curves of the five electrodes all show a pair of obvious redox peaks. However, the peak area of Fe3O4/MoS2/NiS (5%) is larger than other materials. From the cycle capacity diagram (Figure 5), the impedance data (Figure 7), the XRD (Figure1 and Figure S2) and XPS (Figure S1 and Figure S3) results before and after cycling, it can be concluded that MoS2 and NiS additives can not only improve the discharge capacities and charge transfer resistance, but also optimize the structural stability. The experimental results show that the MoS2 and NiS additives can effectively eliminate the passivation phenomena in iron electrodes, reduce the electrode polarization and increase the reversibility capacity.

  1. Figure 8 caption is not clear.

Response: We are sorry for this error. It has been corrected as “Cycling performance and charging-discharging curves of three electrodes at current densities of 600 mA g-1(a,b), 1200 mA g-1(c,d), and 2400 mA g-1(e,f).”.

  1. Authors should be provided the EDX and XPS results for the compositional confirmation.

Figure S1. XPS results of Fe3O4/MoS2/NiS (5%). (a) Wide-range scanning result of XPS spectrum. High-resolution XPS spectrum of Fe 2p (b), O 1s (c), Mo 3d (d), S 2p (e) and Ni 2p (f).

Response: Thanks for your suggestion. The EDX results of Fe3O4/MoS2/NiS are shown in Figure 3. And the XPS results of Fe3O4/MoS2/NiS are shown in Figure S1 in the supporting information. As shown in Figure S1a, the MoS2 and NiS were successfully added in the Fe3O4 anode. The 2p3/2 peaks of Fe(III) and Fe(II) at 714.08 eV and 712.61 eV, and the corresponding 2p1/2 peaks at 727.56 eV and 725.72 eV are considered as the specific spectrum of Fe3O4 (Figure S1b) (Wang, F. et al. Scientific Reports 2017, 7, 6960.). The binding energy (BE) values for the most intense Mo 3d (Figure S1d) and S 2p (Figure S1e) doublets were found to be 230.80 eV (Mo 3d5/2) and 162.35 eV (S 2p3/2). These values correspond to MoS2 peaks reported in previous work (Baker, M. A. et al. Applied Surface Science 1999, 150, 255−262). As shown in Figure S1f, the binding energy at 857.1 and 874.65 eV represent the existence of Ni2+, which agrees well with the reported literature (Guan, B. et al. Chemical Engineering Journal 2017, 308, 1165-1173.). The XPS test results further show that MoS2 and NiS additives were successfully added to Fe3O4 anode. It has been complemented in the manuscript.

  1. How authors confirmed the hindrance of Fe3O4passivation after the inclusion of MoS2 and NiS?

Response: Thanks for the referee’s comments. Iron electrodes are usually composed of iron tetroxide, conductive agent and binders. Fe3O4 is reduced to iron particles in the charge process. While in the discharge process, iron particles lose electrons, combined with OH- and formed as Fe(OH)2 at the electrode/electrolyte interface.  Therefore iron electrodes are easily passivated due to the formation of the iron-hydroxide, leading to a low discharge capacity and poor charge-discharge performance at low temperatures and large rates. In previous studies, sulfide additives have significant effects, inhibiting the passivation of iron electrodes and increasing the utilization rate of the active materials. (Posada, J.O.G. et al. Journal of Power Sources 2014, 268, 810–815; Posada, J.O.G. et al. Journal of Power Sources 2014, 262, 263–269.). The electrochemical tests and structural analysis in this paper also show that the MoS2 and NiS additives can effectively eliminate the passivation phenomena in iron electrodes, reduce the electrode polarization and increase the reversibility capacity.

  1. Authors are suggested to providethe structural and/or surface analysis after the stability test.

Figure S2. XRD patterns of the Fe3O4/MoS2/NiS (5%) sample before cycle and after 100 cycles.

Figure S3. XPS results of Fe3O4/MoS2/NiS (5%) samples after 100 cycles. (a) Wide-range scanning result of XPS spectrum. High-resolution XPS spectrum of Fe 2p (b), O 1s (c), Mo 3d (d), S 2p (e) and Ni 2p (f).

Response: We thank the referee’s suggestion. The XRD and XPS results of Fe3O4/MoS2/NiS after cycling are provided in the supplementary information (Figure S2 and Figure S3). Compared with the primary Fe3O4/MoS2/NiS (5%), the XRD peaks of the electrode after 100 cycles correspond to nickel foam and MoS2, respectively. No other new peaks are found, indicating that the crystal structure is stable after cycling. The full and high-resolution XPS spectrum also indicate that the composition of Fe3O4/MoS2/NiS (5%) electrode after cycling has not changed obviously. Therefore, both XRD and XPS results of the materials after cycling indicate that the sulfide additives can reduce the electrode polarization and improve the cycling stability. It has been complemented in the manuscript.

  1. Title of the manuscript is not clear. Should be updated the title appropriately.

Response: According to your kind suggestion, the title was updated as "The synergistic effect of MoS2 and NiS on the electrical properties of iron anodes for Ni-Fe batteries". It has been revised in the manuscript.

Reviewer 3 Report

I believe that this paper provides interesting results for future research and deserves to be published in Nanomaterials. However, there are some observations that the authors must consider before the manuscript is published:

-The authors must specify exactly the synthesis/processing procedures. For example, the authors use "a certain amount of" and must specify the exact amounts so that the experiments can be repeated by other researchers.

-The authors indicate that MoS2 was provided by Henan Kelong Company, but do not indicate whether this material was exfoliated or whether it was used as received. From the SEM images in Figure 2, I understand that the MoS2 is not exfoliated. Do the authors consider that this fact is not relevant for the behavior of the electrode? If so, they should indicate this in the revised version of the manuscript.

-In Figure 3, the authors indicate that it corresponds to "Photoelectron spectroscopy of the Fe3O4/MoS2 (5%)/NiS (5%) sample." I guess this is a bug, and they need to fix it.

Author Response

I believe that this paper provides interesting results for future research and deserves to be published in Nanomaterials. However, there are some observations that the authors must consider before the manuscript is published:

The authors must specify exactly the synthesis/processing procedures. For example, the authors use "a certain amount of" and must specify the exact amounts so that the experiments can be repeated by other researchers.

Response: Thank you so much for your suggestion. The synthesis/processing procedures were specified in the revised manuscript. “The reagents used in the experiments were all analytical grade and were used without further purification. Fe3O4 was synthesized by chemical coprecipitation. The specific steps were as follows: 27.8 g of Fe3O4·6H2O (97%, Aladdin) and 7.5g of NH4NO3 (98%, Macklin) were dissolved in 75 mL deionized water, and then the mixture was transferred to a three-necked flask. The reaction flask was placed in a water bath at 95 oC, 10 ml of 2 mol/L NaOH (95%, Aladdin) solution was added dropwise to the above solution and under continuous magnetic stirring, and the solution was continually stirred for 30 min after the dripping was finished.”

The authors indicate that MoS2 was provided by Henan Kelong Company, but do not indicate whether this material was exfoliated or whether it was used as received. From the SEM images in Figure 2, I understand that the MoS2 is not exfoliated. Do the authors consider that this fact is not relevant for the behavior of the electrode? If so, they should indicate this in the revised version of the manuscript.

Response: Thank you so much for your suggestion. The MoS2 sample provided by Henan Kelong Company was used as received without exfoliation. And the exfoliation treatment of MoS2 is not relevant for the behavior of the electrode in this work. It has been revised in the manuscript.

In Figure 3, the authors indicate that it corresponds to "Photoelectron spectroscopy of the Fe3O4/MoS2 (5%)/NiS (5%) sample." I guess this is a bug, and they need to fix it.

Response: Thank you so much for your suggestion. We are sorry for the error. The caption of Figure 3 was revised as “EDX images of the Fe3O4/MoS2/NiS (5%) powder (a) and the electrode sheet (b)”.We have corrected the typos throughout the whole manuscript as far as we can.

Round 2

Reviewer 2 Report

Revised manuscript is suitable for publication.